# Into the Tissues: Extracellular Matrix and Its Artificial Substitutes: Cell Signalling Mechanisms

**DOI:** 10.3390/cells11050914

**Published:** 2022-03-07

**Authors:** Aleksandra Bandzerewicz, Agnieszka Gadomska-Gajadhur

**Affiliations:** Faculty of Chemistry, Warsaw University of Technology, Noakowskiego 3 Street, 00-664 Warsaw, Poland; aleksandra.bandzerewicz.stud@pw.edu.pl

**Keywords:** extracellular matrix, cellular receptors, cell adhesion, cell signalling, scaffolds, biomaterials

## Abstract

The existence of orderly structures, such as tissues and organs is made possible by cell adhesion, i.e., the process by which cells attach to neighbouring cells and a supporting substance in the form of the extracellular matrix. The extracellular matrix is a three-dimensional structure composed of collagens, elastin, and various proteoglycans and glycoproteins. It is a storehouse for multiple signalling factors. Cells are informed of their correct connection to the matrix via receptors. Tissue disruption often prevents the natural reconstitution of the matrix. The use of appropriate implants is then required. This review is a compilation of crucial information on the structural and functional features of the extracellular matrix and the complex mechanisms of cell–cell connectivity. The possibilities of regenerating damaged tissues using an artificial matrix substitute are described, detailing the host response to the implant. An important issue is the surface properties of such an implant and the possibilities of their modification.

## 1. Introduction

A cell is the smallest structural and functional unit of a living organism, capable of carrying out all basic life processes. Cells show remarkable morphological and biochemical diversity. They may constitute an independent organism or hierarchically build multicellular organisms. Appropriate structural support makes the orderly organisation of cells into tissues and organs possible. The substance filling the spaces between cells is called the extracellular matrix (ECM). It is a network of proteins and polysaccharides secreted locally by the cells. The composition of the extracellular matrix determines the properties of the tissue it builds [1].

The correct development of tissue and the maintenance of its functionality result from a controlled flow of information between cells. Receiving and responding to signals from the surrounding environment is made possible by receptors. Receptors process and amplify signals and transduce them into the cell via a series of signalling molecules. Coordinated interaction between cytoskeleton, receptor and matrix is essential. Therefore, cell adhesion is a dynamic process and not just a passive anchoring to the matrix [2,3,4].

As a result of mechanical damage or lesions, the structural integrity of the tissue can be damaged. The body is often incapable of naturally repairing such defects. Therefore, the cells need an artificial scaffold that mimics the natural extracellular matrix and the properties of the regenerated tissue. Such an implant should accelerate the healing process and, above all, not provoke a negative response from the body. It depends mainly on the surface properties of the implant. Both new materials and methods of their modification are constantly being sought to improve patient treatment [5].

## 2. The Extracellular Matrix—Composition, Structure, Functions

The extracellular matrix is often referred to as the natural scaffold of tissues and organs. Still, the functions of this structure go far beyond being mere physical support for the cells. The extracellular matrix regulates cell life processes from adhesion, differentiation, proliferation, migration to apoptosis because of the extensive network of matrix components, their ability to interact with each other, with signalling factors and with membrane receptors [2,6,7,8].

The ECM is an essential component primarily of connective tissue, one of the four main tissue types in the human body (along with epithelial, muscle, and nerve tissue). It is a complex mixture of water, proteins, and polysaccharides. The balance of these three components is determined mainly by the tissue type (cartilage, bone, fat, connective tissue that builds tendons, etc.) and by its development stage and pathophysiological state [2,9,10,11,12]. The ECM components are locally synthesised and secreted by cells, mainly fibroblasts, the most numerous, although least specialised, of the connective tissue cells [8,13,14]. The organisation of the matrix structure is influenced by the arrangement and orientation of the intracellular cytoskeleton [8].

### 2.1. Two Types of the Extracellular Matrix

Although the basic organisation of the ECM structure is the same throughout, two basic types of the matrix are distinguished by their location and composition: the interstitial matrix, which forms a three-dimensional porous network surrounding the cells (especially connective tissues), and the pericellular matrix, which is more compact and forms a layer adjacent to the cells [15,16] (see Figure 1).

The interstitial matrix can be equated with the “proper” matrix, as it forms the structural scaffolding for the cells. Its basic components are heterotypic fibrils, composed mainly of type I collagen with small amounts of type III and V collagens in variable proportions, both playing an important role in fibrillogenesis [16]. The collagens of the interstitial matrix are mostly secreted by fibroblasts [17]. Important components of this “amorphous three-dimensional gel” also include fibronectin and elastin, involved in the organisation of the structure [18,19].

A typical example of the pericellular matrix is the basement membrane, a delicate and flexible nanostructure that separates the epithelium from the deeper layers of connective tissues. It ensheathes smooth, skeletal, and cardiac muscle fibres, Schwann cells, and adipocytes. The basement membrane forms a specific boundary of many organs in mature tissues, often surrounding their functional units [16,20,21,22]. It is mainly composed of type IV collagen, laminins, nidogens and heparan sulfate proteoglycans (HSPGs): perlecan and agrin [21]. The basement membrane contains so-called matricellular proteins that do not contribute to its physical stability or structural integrity, although they may be connected to building components. Instead, they have regulatory functions and interact with surface receptors, proteases, hormones or other biologically active molecules. They may be tissue-specific in terms of function and structure [23,24,25,26]. Matricellular proteins include SPARC (secreted protein acidic and rich in cysteine, or osteonectin; characteristic of mineralising tissues, mainly bone), thrombospondin-1 (which is rich in platelet α-granules; when secreted, it causes, among other things, activation of TGF-β1, i.e., transforming growth factor-beta 1), and tenascin-C (the gene of this protein is expressed during embryonic life, while in adult tissues, tenascin-C is very poorly detectable, being present rather in the course of pathological processes [27,28,29]. The tasks of the basement membrane include regulation of tissue development, function, and regeneration by controlling the cellular response. It is a storehouse of growth factors and modulates their activity and concentration. It serves to maintain the phenotype of the cells it surrounds [30]. The interstitial matrix and the basement membrane are closely interconnected, ensuring the integrity of the tissue [16].

The functional equivalent of the basement membrane described above is a type of pericellular matrix that surrounds chondrocytes in articular cartilage [31]. It acts as a physical barrier that filters molecules entering and leaving the cells. Together with an adjacent thin layer of matrix, each chondrocyte forms a structural unit called a chondron [32]. The morphology of chondrons varies. They can take a discoid/ellipsoid/rounded shape and a variable orientation, which depends on the position, i.e., the depth of location in the cartilage. In some cases, a chondron comprises more than one cell (up to four) [33]. In this case, an essential component of the pericellular matrix is type VI collagen, although it generally constitutes a negligible percentage of the collagens of cartilage tissue [34]. However, because of its specific presence in the chondrocyte environment of articular cartilage, it often serves as a marker of chondrons [34,35]. A characteristic feature of articular cartilage is the small number of chondrocytes compared to the extensive extracellular (interstitial) matrix for which synthesis, organisation, and maintenance they are responsible [34].

When describing the types of the ECM, the term ‘pericellular matrix’ is sometimes omitted and replaced by the basement membrane itself, leading to the mistaken assumption that it is the same structure [17,18,19]. However, the basement membrane should be considered a more specialised form of the pericellular matrix.

### 2.2. Major Components of the Extracellular Matrix and Their Functions

#### 2.2.1. Collagens

Collagen proteins account for up to 30% of all proteins in vertebrates and are major extracellular matrix components. The basic collagen macromolecules are composed of three same (homotrimers) or different (heterotrimers) polypeptide chains. They are characterised by the repetitive Gly-X-Y sequences, where X usually stands for proline and Y for 4-hydroxyproline. The intertwined chains form a specific triple helix structure [17,36,37].

Due to their supramolecular organisation, fibrillar (types I, II, III, V, XI, XXIV, and XXVII) and non-fibrillar collagens are distinguished. Characteristic for the non-fibrillar collagens is a disrupted continuity of the typical structure. Compared to fibrillar collagens, they contain shorter (although more numerous) helical (collagenous) domains interspersed with so-called telopeptides, i.e., non-helical domains. As a result, they may occur in various forms, forming, e.g., network systems (types IV, VIII, X), anchor fibres (type VII), beaded filaments (type VI), or belong to the FACIT group (i.e., fibril-associated collagen with interrupted triple helix, types XI, XII, XIV, XVI, XIX-XXII). The terminology and affiliation are not fully systematised. Collectively, collagens form a family of 28 proteins [38,39,40,41,42,43,44].

Historically, collagens were thought to have only a supportive function. Although their main function is indeed to form the structural scaffolding of cells (especially for types I, II, and III), it is known that their role is much broader [15,18,37,45]. Collagens are involved in regulating the course of cell adhesion (as ligands of cell receptors) [46,47,48,49], cell migration (contact guidance) [50,51,52] and tissue reconstruction and remodelling [37,40,45,53,54]. Not only is the physical deposition or movement of cells itself important, but also the processes conditioned by this, e.g., wound healing, immune response, etc. Although collagens are present in most body tissues and affect their mechanical properties, their distribution varies, e.g., type I collagen is characteristic of bone, skin and tendon, and type II collagen of cartilage tissue [55,56].

#### 2.2.2. Elastin

Elastin is a hydrophobic fibrillar protein, which owes its characteristic elastic properties to extensive covalent cross-linking of the structure [57]. The monomer from which the mature insoluble protein is formed is tropoelastin, secreted by fibroblasts, smooth muscle cells, endothelial cells, respiratory epithelial cells, chondrocytes, and keratinocytes [58,59,60,61,62]. After secretion into the intercellular space, tropoelastin spontaneously associates into larger particles through interactions between hydrophobic domains in a process called coacervation [63]. Such precursors undergo oxidative deamination of lysine residues in tropoelastin. The process is catalysed by LOX family enzymes (lysyl oxidases). The result is the formation of allysine from lysine. Cross-linking occurs via the reaction between lysine and allysine residues (Schiff base reaction) or by aldol condensation of two allysine residues [64,65,66,67]. The final fibres are not composed of elastin alone. Elastin forms a core (about 90% of the whole structure), covered by an envelope of microfibrils composed mainly of glycoproteins from the fibrillin group (fibrillin-1 and -2) [58,68,69]. In this way, elastic fibres are formed, giving tissues susceptibility to stretching. They are a particularly important component of blood vessel walls, skin, lungs, heart, tendons, ligaments, bladder, elastic cartilage tissue (e.g., auricle, larynx, epiglottis), etc. [15,68,70].

The gene expression and formation of elastic fibres occur at early development stages —prenatal and early childhood. De novo production of elastin in adult organisms is unlikely to occur, which is quite uncommon among the ECM components [58,71,72,73,74]. However, elastin has high metabolic stability and a half-life of approximately 70 years, making the limited synthesis time sufficient (by comparison, the half-life of type VII collagen is estimated to be approximately one month [75,76]. The adult organism cannot reconstitute elastic fibres that become damaged or degrade progressively with age. They are then repaired incorrectly and consequently do not perform their normal functions. The tissues become too stiff, leading to cardiovascular disease, lung disease or typical signs of ageing, such as loss of skin elasticity [15,67,77,78].

#### 2.2.3. Proteoglycans

Proteoglycans are macromolecules of a complex three-dimensional structure. They are composed of a protein core covalently linked to one or more chains of glycosaminoglycans (GAGs), a type of linear, unbranched heteropolysaccharides. The glycosaminoglycan chains may belong to one or different types. Based on localisation, four basic groups of proteoglycans can be distinguished: intracellular and those occurring on the cell surface, in the pericellular space (basement membrane) or intercellular space [15,79,80,81].

GAG chains are built by repeating disaccharide units, where one residue is an amino sugar (N-acetylated hexosamine), and the other is uronic acid (D-glucuronic or L-iduronic acid). GAGs differ in the type of monosaccharide residues and the geometry of the linkages between the constituent units (α- and β-glycosidic linkages) and the degree of sulfation of the polysaccharide backbone and the position of this substitution. Based on the chemical structure of the chain, four basic groups of glycosaminoglycans are distinguished: heparan/heparan sulfate, keratan sulfate, chondroitin sulfate/dermatan sulfate, and hyaluronic acid [15,82,83,84,85,86]. Hyaluronic acid represents the simplest type of structure. It is the only one that does not contain sulfate groups (hydroxyl groups are not esterified with sulfate groups) and does not undergo complex modifications in the Golgi apparatus [84,87,88]. Unlike other GAGs, it does not form covalent bonds with proteins and, therefore, is not part of typical proteoglycans. Instead, it can exist in the form of non-covalent complexes with other protein components of the ECM [85,86,89,90,91].

Hyaluronan has excellent water retention ability. It is abundant in the skin, cartilage, brain, vitreous body, umbilical cord, and synovial fluid. Its physical and physiological properties depend on molecular weight and concentration in the tissue. When highly concentrated, hyaluronan molecules form a three-dimensional meshwork structure exhibiting remarkable viscoelasticity. The organised structure acts as a molecular sieve of proteins and other macromolecules. Hyaluronan is reported to modulate cellular behaviours via the reprogramming of cellular metabolism coupled to its production [92]. Hyaluronan activates signalling cascade by interacting with CD44 receptor. CD44 was originally identified as a hyaluronan and hyaluronic acid receptor but can bind to various other ligands. It also serves as a marker for stem cells of several types [93].

Glycosaminoglycan chains (and, therefore, the proteoglycans) are negatively charged. It is the result of carboxyl and sulfate residues in their structure [88,94]. Due to the strong negative charge, these molecules tend to elongate in solution under physiological conditions. This allows them to bind large amounts of water and form a gel. Such properties provide tissues with resistance to deformation by high physical forces, as exemplified by aggrecan, the most important cartilage proteoglycan [79,86,95,96]. The proteoglycan family also includes compounds, such as syndecans (trans-membrane receptors; they bind numerous ligands present in the ECM, mediate signal transduction, cell adhesion, migration et al.) [97,98], serglycin (the only known intracellular proteoglycan; found in leukocyte granules, regulates granulopoiesis) [99,100,101], perlecan and agrin (characteristic of the basement membrane, regulators of many cellular processes; agrin is involved in the formation of neuromuscular synapses) [102,103,104,105,106,107] and fibromodulin (involved in the collagen fibrillogenesis) [108,109].

#### 2.2.4. Glycoproteins

Like proteoglycans, glycoproteins are composed of covalently linked protein and carbohydrate parts. However, the saccharide chains are much shorter, contain no (or few) repeating units, and are usually branched [2,110,111]. Glycoproteins often act as connectors in the ECM, as they have functional groups capable of binding other proteins, growth factors, or receptors [2,112,113]. Their participation is essential for many biological processes: fertilisation, immune and inflammatory response, blood coagulation, wound healing, etc. [112,114,115,116,117,118,119,120]. The two most important glycoproteins are fibronectin and laminin. The glycoprotein family also includes fibulins [121], tenascin [122], fibrinogen [123], vitronectin [124], osteonectin [27], bone sialoprotein [125], and reelin [126].

The basic structural unit of fibronectin is a dimer composed of two nearly identical polypeptide chains linked by a pair of disulfide bonds. Each such chain is built by irregularly repeating amino acid units (types I, II, and III), forming a mosaic structure of the protein. The molecules consist of domains, i.e., differently structured sections with different functions [127,128,129]. Fibronectin contains domains capable of interacting with the ECM proteins (e.g., collagen), glycosaminoglycans, surface receptors and other fibronectin molecules. Due to these properties, fibronectin can simultaneously bind to cells and components of the surrounding matrix [128,130,131,132,133,134]. In the body, fibronectin exists in two forms: soluble plasma fibronectin (synthesised by hepatocytes and secreted into the blood) and insoluble cellular fibronectin (produced by fibroblasts, endothelial cells, chondrocytes, myocytes, and others). The insoluble form is a fibrillar cross-linked structure on the cell surface and in the ECM. It is responsible for cell adhesion, proliferation, migration, and the ECM protein deposition [128,135,136,137,138,139]. Both forms of fibronectin are encoded by one gene, while structural differences result from alternative mRNA splicing [140,141].

Laminins are a group of large, multi-domain glycoproteins of a heterotrimeric structure. The three subunits (α, β, and γ chains) connected by a pair of disulfide bonds form a characteristic Latin cross-shaped structure (a Y-shape/rod shape form is also possible [142,143,144]). The three shorter arms (their globular N-terminal domains) are mainly involved in laminin polymerisation and network self-assembly. At the same time, the longer one mediates cell–cell interactions by binding to receptors [145,146,147,148,149]. Proteins of the laminin family are an integral part of the basement membrane and play an essential role in forming and maintaining its structure. A critical step in developing the basement membrane is the polymerisation of laminin [149,150,151]. This process is initiated by binding laminin molecules to the cell surface. A connection is formed between the long arm of the protein and the receptors—cognate integrin and dystroglycan. As a result, there is a local increase in the concentration of laminin, and after exceeding a critical value, polymerisation occurs. The structure, thus, formed binds to nidogens and HSPGs (perlecan). The entire network is further stabilised by polymerising type IV collagen [151,152,153,154,155,156,157,158]. The basement membrane layer built up by the complex network of the described components is called lamina densa (the middle layer between the lamina lucida and the lamina fibroreticularis [159].

### 2.3. The Dynamic Structure of the Extracellular Matrix

The structure of the extracellular matrix undergoes continuous remodelling, during which changes in its composition and overall architecture occur. Cells embedded in the ECM are actively involved in its reorganisation. In addition to synthesising and secreting building components, they are also the source of enzymes that degrade these components. Remodelling processes are complex and must be tightly regulated to maintain environmental homeostasis [19,160,161,162].

Protein-degrading enzymes belong to the class of hydrolases and are called proteases (proteinases). Depending on the mechanism of catalysis, they can be divided into several families, including serine proteases (serine residue in the enzyme active site), cysteine proteases (cysteine residue) or metalloproteases (they require the presence of a metal cation in the active centre). These enzymes can be secreted by the cell into its external environment or remain anchored in the cell membrane [163,164].

The main group of enzymes involved in ECM degradation are the zinc-dependent matrix metalloproteinases (MMPs). More than 20 representatives of this group are known, capable of degrading different types of collagen, gelatin, elastin, laminin, fibronectin and many others [165,166,167]. The sources of MMPs are mainly connective tissue cells (fibroblasts, osteoblasts), inflammatory cells (macrophages, neutrophils, mast cells), and endothelial cells [165,168]. MMPs are secreted in the form of zymogens, inactive precursors that must undergo biochemical modifications to be activated [19,165,168]. Through controlled degradation of ECM proteins, metalloproteinases facilitate cell migration and trigger the release of growth factors [169,170,171]. They participate in tissue remodelling, an interesting example of which is postpartum uterine involution. In addition, they regulate angiogenesis (blood vessel formation), wound healing, embryonic development, etc. [165,172,173]. In pathological states, their abnormal and/or increased activity contributes to the course of cardiovascular, cancer, autoimmune diseases, etc. [165,174,175,176].

The proteolysis occurring in tissues relates not only to the extracellular matrix per se but also concerns the so-called ectodomain shedding, i.e., proteolytic cleavage of cell surface proteins. Modification, degradation, and changes in the activity of these proteins are one of the mechanisms of the cell’s response to changes in microenvironment conditions [177,178]. Enzymes of the ADAM (a disintegrin and metalloproteases) family, also known as adamalysins, are mainly involved in this process. They have various functions, primarily engaged in intercellular interactions and signal transduction [19,179,180]. The release of biologically active extracellular domains of multiple proteins (cytokines, adhesion molecules, growth factors) from the cell membrane can contribute, e.g., to inflammation (physiological and pathological), as occurs as a result of ADAM17 enzyme activity. The pro-inflammatory action of this sheddase consists of a modification of the cell surface and enrichment of its environment with active soluble molecules [181,182,183,184]. The structure and function of ADAM group proteins are similar to the metalloproteinases found in snake venom, responsible for the typical effects of snakebites (haemorrhage, tissue necrosis) [185].

### 2.4. The Extracellular Matrix as a Storehouse of Growth Factors

The ECM significantly influences the cell’s most important natural biological processes: growth, proliferation, and programmed death [186]. In addition to mediating interactions and activating relevant mechanisms by contact with its building proteins, the ECM serves as a storehouse of growth factors (and proteases and protease inhibitors). These molecules can be released by proteolytic degradation of the matrix, and the degradation itself regulates the rate, site and intensity of such activation. The fact that growth factors are stored in the vicinity of cells favours increased specificity of their action [19,187,188,189].

Growth factors are generally not freely dispersed in the extracellular space but bind, for example, to heparan sulphate proteoglycans. HSPGs then participate in the matrix storage function by preventing the movement and proteolysis of growth factors. They allow their controlled release when necessary. However, another role of HSPGs is also to bind to such molecules to activate them. Then, they act as a coreceptor in ligand–receptor interactions [187,190,191,192]. The type of interaction of HSPGs with growth factors depends on the localisation of these proteoglycans. They may remain anchored to the cell membrane or form a structural component of the ECM [193].

A well-studied group is the fibroblast growth factors (FGF), which include 22 proteins with key functions in cell development, morphogenesis, tissue repair processes, and angiogenesis. They are among the neurotrophic factors, i.e., those that stimulate and regulate neurogenesis. Some are being investigated for involvement in the development of depression [192,194]. FGF molecules are mainly bound by heparan sulfate and heparin chains [195,196]. Proteolytic release of FGF allows subsequent binding of FGF ligands to receptors on the cell surface. This stimulates cell signalling [2].

Another example is transforming growth factor-beta (TGF-β), specifically its three isoforms, responsible for stimulating and inhibiting cellular proliferation. Among other things, the TGF-β cytokine controls the course of wound healing by interacting with different cell types. For example, TGF-β1 is released in large amounts at the wounding site by platelets and stimulates chemotaxis of monocytes and fibroblasts [197]. Cells secrete biologically inactive TGF-β molecules, which, in the latent form, are bound by matrix proteins (via glycoproteins of the LTBP family bound to ECM proteins, mainly fibronectin and fibrillins) in the form of complexes [188,198,199]. TGF-β activation (in vivo and in vitro) can occur in several ways: through enzymatic proteolysis of the complex (matrix metalloproteinases, serine proteases) [200,201], interaction with integrins [202,203,204] or other proteins [205,206] and in response to physicochemical conditions (radiation [207,208], low/high pH [209,210], temperature [211], and reactive oxygen species [212]).

### 2.5. Anoikis—Programmed Death

The normal functioning of most cells depends on their proper connection to the matrix. The cell is informed of this connection mainly by integrins (one type of membrane protein), acting as the ECM signal transducers [3,8,213,214]. However, if a cell detaches from the surface of the matrix, there is a risk that it will move and become embedded elsewhere. To prevent the abnormal proliferation of cells away from their parent tissue, a defence mechanism called anoikis (Greek for homeless), a type of apoptosis, is induced in the adherent cell if contact with the matrix is lost [215,216,217,218].

The term anoikis was introduced in 1994 by Frisch and Francis, who conducted studies on Madin–Darby Canine Kidney (MDCK) [217]. The relationship between a cell’s ability to anchor to a substrate and proliferate by affecting cell cycle progression was already known [219]. In subsequent years, this form of apoptosis was studied and described in a number of different types of adherent cells, such as fibroblasts [220,221], endothelial cells [222,223], keratinocytes [224,225], oligodendrocytes (glial cells) [226], and neurons (dopaminergic) [227], as well as bronchial [228,229], intestinal [230,231,232], or mammary gland epithelial cells [233]. The mechanism of anoikis can be induced by various signalling pathways, all of which ultimately lead to the activation of proteolytic enzymes of the caspase family and the degradation of cellular proteins [224,233,234,235,236]. Apoptotic cell death comes to an end with removing the cell’s genetic material, i.e., DNA fragmentation controlled by endonuclease enzymes [236,237].

The acquisition of resistance to anoikis is a characteristic feature of circulating tumour cells, enabling them to survive in non-adherent conditions. After detaching from the primary tumour, they are transported with the peripheral blood and are responsible for forming metastases [236]. Studies suggest that there are different mechanisms for the development of such immunity [238,239,240,241].

### 2.6. Genetic Mutations of Matrix Components and Their Consequences

To emphasise how important the extracellular matrix components are for the proper functioning of the organism, one should look at the consequences of abnormalities in their synthesis. The consequence of mutations in genes encoding the ECM proteins is a wide group of genetic disorders of connective tissue, with better or worse understood pathogenesis and often varied course. Dysfunction of the matrix may occur due to two different mechanisms of mutations. The first one involves a violation of the structural integrity of the matrix by quantitative reduction in its components as a result of nonsense mutations (formation of a premature stop codon) and/or frameshift mutations (insertion/deletion of a number of nucleotides indivisible by three). In the second type, the secretion of mutant proteins qualitatively affects the matrix structure, as it disrupts the stability of their interactions with normal, genetically unaltered components [242,243].

One of the most important is osteogenesis imperfecta (OI), a group of inherited disorders characterised by low bone mass leading to increased bone susceptibility to fracture [242,243]. The incidence rate is estimated to be around 1/10,000 births [244], so it is a relatively rare condition. In most cases, OI is caused by mutations in the genes encoding the α1(I) and α2(I) chains of type I collagen, manifesting as reduced production of this protein or its structural deformities [244,245]. Several clinical types of OI have been identified. According to Sillence’s 1979 classification, four are distinguished based on clinical and radiological symptoms and mode of inheritance [246]. New OI types have been described in recent years, resulting from mutations in so-called non-collagenous genes [247]. Clinically, however, they do not differ from the classical forms of this disease and are, therefore, included in them [242]. A rather peculiar symptom of OI is the blue sclera. Collagen fibres are one of the main building components of the sclera. A reduction in their thickness causes the deeper-laying choroid to become visible [244,248,249].

Another group of diseases mainly associated with abnormalities in the synthesis of fibrillar collagens or enzymes responsible for their post-translational processing is Ehlers-Danlos syndrome (EDS). Thirteen subtypes of EDS have been recognised (six according to the older Villefranche classification), manifested by a range of symptoms. The most characteristic is joint hypermobility, skin hyperelasticity, and general tissue tenderness [250,251,252]. The changes seen in classical EDS include loosely and irregularly packed collagen fibres and fibres called “cauliflowers” because of their characteristic cross-sectional shape. In the normal fibres, the cross-section is circular [251,253]. People affected by the vascular type of Ehlers–Danlos syndrome have translucent skin, a distinctive facial appearance (thin lips and nose, small chin, large eyes), and are prone to spontaneous bruising and a rupture of the arteries, intestine, and, in the case of pregnancy, the uterus [254,255]. It is estimated that between 1/2500 and 1/5000 people suffer from EDS. These numbers may be underestimated because patients with mild symptoms often go undiagnosed [252,255,256]. Depending on the type, EDS is inherited in an autosomal dominant or recessive manner, but a de novo mutation may also occur [252]. Like OI, Ehlers–Danlos syndrome is an incurable disease. Current therapies are aimed only at improving the quality of the patient life [255].

Genetic diseases of connective tissue do not only include abnormalities in collagen synthesis. One of the representatives of fibrilinopathies is Marfan syndrome (MFS), caused by a mutation in the FBN1 gene located in chromosome 15, which encodes fibrillin-1 [257]. Structural defects of this protein result in a violation of the stability of elastin fibres and their disorganisation in the connective tissue of various organs [258,259]. In addition, fibrillin-1 can bind TGF-β, so its dysfunctions result in increased levels of free TGF-β, activating abnormal degradation mechanisms [260]. The greatest threat to the lives of patients diagnosed with MFS is related to cardiovascular dysfunction. Progressive aortic root dilatation associated with the disintegration of elastin fibres causes aortic dissection, dangerous especially in the ascending part, i.e., closest to the heart. The formation of aneurysms is also possible. If undiagnosed and untreated, such abnormalities can be fatal at an early age [261,262,263]. Nevertheless, the most characteristic changes in the MFS course are those in the osteoarticular system. These include disproportionately long limbs and arachnodactyly (“spider fingers”), deformities of the thorax and spine (scoliosis, pathological kyphosis), protrusion acetabula (medial displacement of the acetabulum into the true pelvis), and overly flexible joints. These changes are often accompanied by ocular abnormalities, such as ectopia lentis [261,264,265]. Due to medical advancement, especially the possibility of preventive aortic aneurysm surgery, the life expectancy of people affected by Marfan syndrome has nearly doubled over the years [261,265,266].

Disorders caused by mutations in genes encoding elements of the extracellular matrix are, of course, far more numerous than those mentioned. They include several other collagenopathies, such as Stickler syndrome [267,268], Bethlem myopathy [269,270], Ullrich congenital muscular dystrophy [271,272], or the dystrophic epidermolysis bullosa [273,274]. Conditions may also result, for example, from abnormalities in the structure and function of perlecan (Schwartz–Jampel syndrome [275,276]), laminin (Pierson syndrome [277,278]), fibulin (age-related macular degeneration [279,280]). Many of those mentioned are rare disorders, still poorly understood.

## 3. Interactions between Cells and Their Environment

Receiving signals from the surrounding environment is essential for normal cell development and function. The environment includes other cells, the extracellular matrix and various soluble factors. Information arriving from outside as a ligand (chemical substance) or physical stimulus is received by a membrane receptor or intracellular receptor. The information is transduced (carried and appropriately transformed) into the cell. It initiates a series of reactions and changes the cell’s physiological behaviour or maintenance of appropriate activity. A complex communication system is the basis for regulating cellular processes and, as a result, the functioning of the whole organism [2,281,282].

### 3.1. Forms of Signalling

The basic type of communication between cells is the transfer of information in the form of a chemical compound, i.e., a protein, peptide, amino acid, lipid or their derivatives. These are various hormones, cytokines, growth factors, neurotransmitters, etc., synthesised by signalling cells. Once released into the intercellular space, they can be bound as ligands by receptors capable of recognising them [283,284,285,286,287,288,289,290]. Depending on the distance the ligand must travel between the signalling and target cell, endocrine, paracrine and autocrine signalling can be distinguished. A fourth type is signalling by direct contact [4,8] (see Figure 2).

Endocrine signalling occurs via hormones. Hormones are produced by specialised cells, secreted into the bloodstream, and distributed throughout the body. With the bloodstream, they reach distant target cells. It is, therefore, long-distance signalling, but it occurs relatively slowly because it depends on the speed of blood flow [4,8,291]. The distance covered by the signalling molecule is shorter in paracrine signalling. It acts as a local carrier and affects cells in the immediate vicinity. A specific form of this type of communication is synaptic signalling. Two nerve cells (or a nerve cell and a target cell) are bound by a connection called a synapse, usually a chemical synapse. There are also electrical synapses, which work by the direct flow of ions. Stimulated by a nerve impulse, the presynaptic (transmitting) neuron releases a neurotransmitter, carrying the information to the postsynaptic (receiving) neuron [4,8,284,292,293,294,295,296]. In autocrine signalling, the cell responds to substances secreted by itself, i.e., it both produces a ligand and has a receptor that binds it. Such a mechanism is fundamental in the early stages of organism development or inflammatory processes. It is also characteristic of cancer cells [4,8,293,297,298,299]. Direct interaction between cells, i.e., juxtacrine signalling, does not use molecules secreted into the extracellular space. Two cells connect via complementary surface proteins, one acting as a signalling agent and the other as a receptor. An example is pathogen recognition by immune cells. Cells can also form gap junctions, water-filled protein channels made up of two connexons, hexameric assembly of connexin proteins. The gap junction is the contact site between the cytoplasm of neighbouring cells. It allows substances, rather small in size, such as calcium ions, to flow between them [4,284,300,301,302,303,304,305].

An important issue is the effect of the ligand binding to the receptor, as not every ligand causes receptor stimulation. A signalling substance that, when bound, changes the receptor’s conformation (i.e., has intrinsic activity) and causes a programmed change in cellular activity is called an agonist. The opposite of an agonist is an antagonist, which has no intrinsic activity despite its ability to bind to a receptor. An antagonist blocks the receptor without eliciting a biological response from the cell and prevents activation by the agonist [306,307].

### 3.2. Receptors

The activity of receptors mainly involves converting one form of signal into another, which usually initiates a multi-step chain of information transfer through several signalling molecules. Often the signal is also amplified. In this way, signalling cascades are formed, leading to an effector response, i.e., various changes in cell activity [308,309,310,311,312,313].

The vast majority of signalling molecules do not enter the cell. The classical model is based on ligand binding by a specific receptor protein. Exceptions include small lipophilic molecules that can cross the barrier formed by the cell membrane, the lipid bilayer. The external signal is then received by intracellular receptors, located mainly in the cell nucleus or cytoplasm [4,8]. Steroid hormones belong to this type of signalling molecules. For years, it has been a common belief that they enter the cell by passive diffusion. However, they may require the involvement of transporter proteins [314,315,316,317,318].

A cell is exposed to hundreds of signalling molecules, so it must respond selectively, i.e., have the right set of receptors. It makes the cell capable of completely bypassing some of the signals. Different cells may respond differently to the same signalling molecule depending on the signalling pathway initiated [319,320,321].

Receptors can be divided into two main groups: intracellular and cell-surface (membrane) receptors. Intracellular receptors have already been described as interacting with steroid hormones. They also bind thyroid hormone, retinoic acid (a derivative of vitamin A) and vitamin D. The receptors for these compounds belong to the nuclear hormone receptor superfamily, formed by structurally homologous proteins. The name can be misleading, as the subcellular location of the unliganded receptors varies. However, after binding to a ligand, they are mainly translocated to the cell nucleus, where they act as transcription factors, i.e., regulate gene transcription. A specific group of orphan intracellular receptors is included in this superfamily. The existence of endogenous ligands binding to them has not been confirmed [322,323,324,325,326].

Cell-surface receptors are anchored to the cell membrane. Depending on the type of information transfer, three subtypes of cell-surface receptors are distinguished (see Figure 3). The first is enzyme-linked receptors (catalytic receptors). The binding of a ligand from the extracellular side causes conformational changes (phosphorylation/autophosphorylation) of the receptor, stimulating enzymatic activity of its cytoplasmic domain. Most commonly, the intracellular domain is responsible for a tyrosine kinase (insulin receptor, growth factor receptors) or serine/threonine protein kinase (TGF-β superfamily receptors) activity. The receptor’s conformation in an inactive form prevents the attachment to the active site (enzymatic domain) of a substrate molecule, i.e., various types of cytosolic proteins that modulate intracellular reactions [327,328,329,330,331,332].

The second group are ligand-gated ion channels, called ionotropic receptors. Ion channels are protein structures that pierce the lipid bilayer and control the flow of ions into or from the cell. A conformational change in the receptor protein caused by the binding of a signalling molecule causes the channel to open. Flow occurs according to a concentration gradient (by diffusion). Ion channels are selective, which means that they distinguish between positive and negative ions. It is determined by the charge accumulated on the side chains of the amino acids. Ion channels are permeable (mainly, but not exclusively) to ions of a given type, e.g., sodium ions [333,334,335,336,337]. The function of channels can be stopped by binding of so-called blockers [338]. In addition to ligand-gated ion channels, there are also voltage-gated ion channels (activated by changes in electrical membrane potential) and stretch-activated ion channels (responding to membrane stress) [339,340,341].

G protein-coupled receptors (GPCRs) are the most numerous and highly diverse cell-surface receptors. All GPCRs are composed of a single polypeptide chain, piercing the cell membrane seven times; hence, they are also called seven-transmembrane receptors (7TM). The hydrophobic transmembrane domains are α-helical regions of the chain, connected by hydrophilic intracellular and extracellular loops. The carboxyl end of the polypeptide (C-terminus) is located on the cytosolic side of the cell membrane, while the amino end (N-terminus) is located in the extracellular region. Together with the extracellular and intracellular loops, they are involved in ligand binding and G-protein interactions, respectively [342,343,344]. G-proteins are proteins binding GTP/GDP likewise. They have a heterotrimeric structure and belong to one of four distinguished families depending on the amino acid sequence in the α subunit [345]. In the inactive state, the α subunit binds GDP and forms a complex with the Gβγ dimer. Activation induced by ligand attachment to the extracellular domain of the GPCR receptor results in the release of GDP, the binding of GTP in its place and the dissociation of Gα-GTP from the βγ subunits. Both structures (Gα-GTP and Gβγ) can participate in further signal transduction to effector proteins. Hydrolysis of GTP to GDP due to the intrinsic GTPase activity of the Gα subunit leads to the re-formation of a Gα-GTP complex with Gβγ, ready for subsequent activation [345,346,347,348,349].

#### Cell Adhesion Molecules (CAMs)

A specific group of proteins are molecules involved in cell–matrix and cell–cell adhesion. The known representatives of this group are classified into four families: integrins, selectins, cadherins, and immunoglobulin superfamily (IgSF) (see Figure 4). This division is due to differences in molecular structure and is strongly related to the heterogeneity of the types of cellular connections formed by these cell-surface receptors [350].

Many CAMs are integrins, heterodimers composed of non-covalently linked α and β units. The large extracellular domain provides the binding site for their ligands, while the much shorter cytoplasmic domain binds to cytoskeletal proteins. Integrins mediate bidirectional signal transmission (environment-cell and cell-environment) as they can be activated by proteins that bind to extracellular and intracellular domains [351,352,353]. Cell-to-cell adhesion is enabled by the formation of connections between integrins and the ECM components: collagens, fibronectin, vitronectin, laminin et al. [354]. In the focal adhesion type connection, the cytoplasmic domain of the integrin is bound to actin filaments through adapter proteins, such as vinculin, paxillin, or talin [355,356,357,358,359]. Focal adhesion kinase, a cytoplasmic tyrosine kinase that, when activated, initiates signalling pathways that regulate various cellular functions, plays an important role in the transmission of the signal received by integrin receptors from the ECM [355,360,361]. Another type of integrin adhesion junction are hemidesmosomes. These specialised multiprotein complexes are responsible for anchoring epithelial cells to the basement membrane by binding to cytoskeletal filaments (keratin intermediate filaments) via plectin. The stability of such connections is vital in maintaining the integrity of the skin, where integrins are involved in the formation of the structure: basal keratinocytes—basement membrane—dermis [354,362,363].

Selectins have a multi-domain structure and are characterised by a lectin domain in their extracellular (N-terminal) part. It allows them to mediate cell–cell interactions by recognising and binding carbohydrates present on cell surfaces. The mechanism is calcium ion-dependent [364,365]. Due to differences in structure and pattern of cell-type expression, three types of selectins are distinguished: leukocyte (L)-selectin, platelet (P)-selectin, and endothelial (E)-selectin. L-selectin is found on the surface of leukocytes. P-selectin is stored in membranes of α granules of platelets and Weibel–Palade bodies of endothelial cells. After activation, P-selectin is incorporated into the membrane of these cells. E-selectin resides in vascular endothelial cells, but a noticeable increase in its surface expression occurs only after stimulation with the appropriate cytokines. Therefore, E-selectins act as a sensitive indicator of the inflammatory process [366,367,368]. The basic function of selectins is to mediate heterotypic interactions between leukocytes and endothelial cells in the initial stages of the inflammatory reaction. The effect of such binding is the so-called leukocyte rolling along the endothelium and their migration to the site of damage/inflammation. It also contributes to the activation of relevant combinations of signalling factors. In the inflammatory process, selectins interact with integrins [367,369,370,371,372,373].

Cadherins are a significant group of calcium ion-dependent CAMs. Primarily, they are mediators of homotypic adhesion, i.e., interaction of two cadherins of the same type. An increase in calcium ion concentration causes a stiffening of the cadherin molecule, which can bind to the cadherin of a neighbouring cell [374,375]. The cadherin family includes several subfamilies. The best known are classical or type I cadherins and atypical or type II cadherins. Representatives of each subfamily differ in the structure of the extracellular part, i.e., in the number of ectodomain (EC) modules with a repetitive amino acid sequence [375,376,377,378]. Classical cadherins are bound to actin filaments of the cytoskeleton via -β-catenin-α-catenin- linkage [379]. It is a dynamic rather than a stable structure, where α-catenin additionally acts as a regulator of the organisation of actin filaments [378,380]. Desmosomal cadherins (desmoglein and desmocollin) have a similar extracellular domain structure to classical cadherins. They participate in cell-to-cell adhesion through structures called desmosomes. The cytoplasmic domain of these cadherins binds to other intracellular anchor proteins (desmoplakin, plakoglobin, plakophilin) and, consequently, to intermediate filaments. One of the hallmarks of the desmosome is the outer dense plaque, consisting of mediatory proteins. Desmosomes provide strong intercellular adhesion. Thus, they are particularly abundant in tissues such as the epidermis and myocardium continually assailed by mechanical forces [378,381,382].

The immunoglobulin superfamily (IgSF) is one of the most numerous and diverse proteins described in the body. The structural feature that determines membership of this superfamily is the presence of one or more characteristic immunoglobulin folds. It is a sandwich structure composed of two opposing antiparallel β-pleated sheets, stabilised by a disulphide bridge [383,384,385]. Ig domains can interact with many types of ligands (integrins, carbohydrates). They readily bind to other Ig domains of the same kind [386]. Due to their properties, many IgSF molecules act as surface receptors (e.g., antigen receptors found on the surface of T cells) or as CAMs [384,386,387,388]. IgSF adhesion molecules often contain some extracellular domains other than Ig, e.g., fibronectin (Fn) type II or III. These are thought to act as ‘fillers’ of the structure, elongating the chain and shifting the position of the Ig-binding domain. Size exclusion mechanism determines the selectivity of the interactions. Fn domains may also be involved in *cis* interactions (between ligand and receptor of the same cell) of Ig molecules, the formation of their clusters on the cell surface and the stabilisation of adhesion [389,390,391]. An important issue related to IgSF molecules is their role in the development and function of the nervous system, where they are involved, among other things, in the processes of axon growth and guidance [392,393,394].

## 4. Artificial Substitutes of the Extracellular Matrix

Cells may be deprived of the necessary connection with the matrix as a result of various types of mechanical or pathological damage. The reconstruction of the structure and restoration of the tissue’s ability to function properly requires replacing the natural extracellular matrix with an artificial substitute. The substitute should support biological regeneration processes. The search for suitable materials and the manufacture of cellular scaffolds for tissue reconstruction is a fundamental goal of tissue engineering. The process is demanding because of the need to match the scaffold’s properties to the tissue’s characteristics. It is essential to confirm that the material is safe for the body and does not cause adverse acute or long-term reactions. Similar requirements are placed on various other implants, not only artificial cellular scaffolds [395,396,397,398,399,400,401,402].

### 4.1. Host Response to Implantation

The body’s first biological reaction to an implant is forming a layer of water on its surface. This happens in just a few nanoseconds. Water molecules form a mono- or bilayer, and the way they are ordered is strongly dependent on surface properties at the atomic level. Water molecules can dissociate on a highly reactive substrate, resulting in the hydroxylation of the implant surface, i.e., it becomes covered with -OH groups. Water molecules can also be strongly bound but not dissociate. Both of these cases occur as a result of contact with a hydrophilic surface. If the surface is hydrophobic, its interactions with water are much weaker. Therefore, the strength of water-binding determines hydrophobicity or hydrophilicity to the surface. It influences the value of the wetting angle formed between the solid and the plane tangent to the droplet deposited on it. Hydrated ions, such as Cl^−^, Na^+^, Ca^2+^, enter the formed water layer [403,404].

Once the aqueous layer covers the material’s surface, proteins from body fluids (extravasated blood/tissue fluid) reach it. In the first stage, mainly smaller proteins with the highest mobility are adsorbed, resulting from faster diffusion of small than large molecules. It is a transient state. A dynamic adsorption–desorption equilibrium is established at the contact surface, as proteins with larger size and a stronger affinity for the implanted material, arriving late, can force the desorption of smaller, weak-bound molecules. This phenomenon is called the Vroman effect. It should be kept in mind that fluids in contact with the implant, such as plasma, contain hundreds of different proteins competing for access to the surface. Therefore, the adsorption–desorption process is much more complex and depends on factors, such as the protein concentration in the fluid. The higher the concentration, the greater the primary surface dominance [404,405].

Proteins usually have an asymmetric structure in which domains of different chemical nature can be distinguished. They have a more or less ellipsoidal shape (globular proteins) [406]. As a result of adsorption, conformational changes of the molecule can occur if it is sufficiently susceptible. It is the effect of binding to the substrate with a privileged side in a given case. As a result, the molecule adopts a certain orientation where part of it invariably contacts the body fluid [407,408,409]. Structurally stable proteins do not readily undergo conformational changes. Their adsorption may occur along the longest axis (“side-on”). Otherwise, this axis is perpendicular to the implant surface (“end-on”) [407]. The issue is not insignificant in the context of establishing a dynamic adsorption–desorption equilibrium, as the ability to structurally reorient increases the possibility of contact with the substrate [407,410].

A major problem with implantation is the foreign body response (FBR), a complex process involving different cell types. Neutrophils are the first to reach the implant site and adhere (via proteins) to the protein-coated surface of the material. Activated neutrophils attempt to degrade the implant by secreting factors, such as proteolytic enzymes or reactive oxygen species. They release chemokines that attract other immune cells, mainly monocytes [411,412,413]. These, in turn, reaching their target, differentiate into macrophages [414]. The number of macrophages at the implantation site increases due to their progressive proliferation. They replace the initial wave of nucleophiles and release further pro-inflammatory factors. It may lead to implant damage and/or the release of toxic substances into the surrounding tissue environment [415,416]. Macrophages may fuse into foreign body giant cells (FBGCs) due to chronic cytokine activity. FGBCs can adhere to the material’s surface for an extended time, leading to collagen deposition and fibrous encapsulation (approximately 3–4 weeks after implantation). As a result, the implant is isolated from the surrounding tissues. It prevents integration and vascularisation and ultimately leads to implant loss [412,417]. The fibrous layer is usually thinner on porous than on solid materials [418,419]. The presence of mast cells, degranulating upon activation, is also characteristic at the implant site. Among other things, histamine is released from the granules. Histamine dilates blood vessels, improves their permeability and facilitates the arrival of other immune cells. Pro- and anti-inflammatory cytokines and angiogenic or profibrotic factors are also secreted [420,421].

Immunosuppressive drugs are used to weaken the body’s immune response and prevent implant rejection. A more recent solution is to incorporate anti-inflammatory agents into the implanted material. They must be released in a controlled manner and at an appropriate rate. An additional requirement is to promote angiogenesis [422,423]. For years, biomaterials engineering has been focused on obtaining biologically inert materials, i.e., minimising the interaction with the organism and reducing the immune response. The contemporary trend is the generation of biomimetic materials, i.e., mimicking the natural solutions of the organism and stimulating the desired responses. These include enhancing or inhibiting the normal functioning of immune cells [424,425,426].

### 4.2. Influence of Material Properties on Cell Adhesion

Cells do not experience direct contact with the implanted material but are only ‘informed’ of its physicochemical properties via proteins deposited on the surface. One of the more important characteristics of the material is the wettability of its surface, which, in the case of an aqueous environment, can be equated with hydrophilicity. It is assumed that the ability of cells to adhere increases on hydrophilic surfaces and decreases on hydrophobic surfaces, even though it is hydrophobic surfaces that are generally considered to be more protein-adsorbent [427].

The surface protein layer that forms shortly after implantation consists mainly of albumin, fibrinogen, immunoglobulin G, fibronectin, vitronectin et al. The first interactions are usually dominated by albumin due to its relatively small size (66 kDa) and high-concentration in plasma [411,428,429]. It binds much more readily to hydrophobic than hydrophilic surfaces but does not promote cell adhesion. The strong adsorption of albumin reduces the likelihood of being replaced by larger adhesion-promoting proteins, such as fibronectin and vitronectin [430,431,432]. The ability of fibronectin to displace surface-bound albumin is limited on hydrophobic surfaces. As a result of the strong binding of albumin molecules, changes in their secondary structure occur and the degree of denaturation increases [433]. Proteins tend to denature as the contact time with the material increases, which occurs when albumin adsorbs onto a hydrophobic material. The binding energy of the adsorbed phase then increases, and, as a result, the probability of desorption decreases [406].

Adsorption occurs more readily if there is a charge difference between the protein molecules and the material surface [434]. Furthermore, the affinity of the protein for the material may show greater specificity than the distinction between hydrophobicity/hydrophilicity and be based on the recognition of specific functional groups [429,433]. Additionally, the cells themselves, depending on the type, show a different preference for the functionality of the surface groups [435,436,437,438].

Adhesion of cells to the implant surface is made possible by integrins recognising and binding to specific amino acid sequences in the polypeptide chain of the adsorbed protein. It mimics the formation of integrin connections with the ECM proteins under natural conditions. The best known among the pro-adhesive sequences is the tripeptide RGD (arginine-glycine-aspartic acid), present, e.g., in the structure of fibronectin [439,440]. One way to modify the material to increase biocompatibility is the coating of tripeptide RGD on its surface in the form of immobilised proteins or short synthetic polypeptide ligands. In addition to RGD, the collagen peptide GFOGER (glycine-phenylalanine-hydroxyproline-glycine-glutamate-arginine) and the laminin-specific sequences IKVAV (isoleucine-lysine-valine-alanine-valine) and YIGSR (tyrosine-isoleucine-glycine-serine-arginine), among others, have been identified [441,442,443,444,445].

Functionalisation of the implant surface with peptides containing the RGD sequence has drawbacks. Integrins that recognise RGD may require the presence of other peptides (synergistic effect) to form a bond. The biological activity of short synthetic peptides is less than that of a whole protein. In turn, modification of these peptides (e.g., by chain elongation) can also result in an undesirable change (increase/reduction) in their activity. Another problem is that cells adhere too strongly to the surface, reducing their movement ability [446,447,448].

An interesting conclusion is provided by the study of cell adhesion on materials exhibiting extreme wettability types. Superhydrophobic surfaces are characterised by a water contact angle value higher than 150°, while superhydrophilic surfaces are around 0°. Although the type of cell determines the contact behaviour, only a few show good adhesion to a surface if the material is superhydrophobic. If the surface has highly hydrophilic and hydrophobic regions, cells will usually selectively attach to the superhydrophilic areas [449].

A significant feature of an implant is the topography of its surface, which, like the chemical composition, influences the interactions with integrins and ultimately stimulates the cellular response [450]. The shape of the natural matrix at the micro- and nanoscale is understood to be the structure formed by the ECM proteins and the neighbouring cells. For synthetic materials, it is the degree of roughness, the type and size of patterns on the surface. Modifications of these features at the nanoscale affect the activity of the adsorbing proteins by forcing specific changes in their conformation. However, the detailed investigation of such relationships is complicated because the cellular response is always a resultant of the influence of different stimuli. In addition, modifications of topography may be accompanied by changes in surface chemistry [451,452,453,454,455].

Techniques to create micropatterns on substrates can be divided into two main types: (1) coating portions of the material with an agent that promotes selective adhesion or (2) applying a layer that blocks adhesion and subsequently removing it without harming the cells embedded around it [456]. The resulting pattern geometry influences the subsequent formation of cells, e.g., it promotes cell elongation. Furthermore, it supports/inhibits the spreading of cells on the surface. It is related to facilitating/hindering their movement, respectively, depending on the continuity of the pattern [457]. The size of the contact area between cells can influence their differentiation, i.e., result in different types of daughter cells [458]. Discontinuities in topography are the cause of local differences in surface free energy. If the cell can detect it, it will modify the contact orientation by reorganising its cytoskeleton. Mechanical signals transmitted to the cell nucleus affect changes at the level of gene transcription and consequently determine cell behaviour. However, the mechanisms underlying the cellular response are still poorly understood [459,460].

In addition to patterns characterised by uniformity of shape and size, cell adhesion is influenced by the surface roughness, understood as the overall three-dimensional topography of the substrate, regardless of its regularity. The surfaces of the used materials are rarely smooth at the molecular level, while roughness is not uniformly describable in this case. Cells must be able to recognise a rough surface to react in a certain way, which is dependent on the cell type, as the primary determining factor is the size of the cell. It means that a cell will recognise a surface as smooth if the peak-to-peak distance is greater than the size of the cell [461,462,463].

Experimental results on the relationship between material roughness and cell behaviour are often contradictory because of different cell types and materials, making it difficult to compare results. However, it is generally accepted that rough surfaces materials promote cell adhesion because they have a larger specific surface area than smooth surface materials. On a smooth surface, the cell needs more connection points to hold on [464,465,466,467,468,469].

## 5. Conclusions

The extracellular matrix is a complex dynamic network structure. Its components are synthesised, secreted, and degraded in a manner controlled by the cells. The matrix fills the spaces between cells, provides structural support, and binds tissues together, providing them with proper mechanical properties. It is an essential component of connective tissues. The intercellular matrix controls cells’ behaviour and vital functions, thus regulating the normal development of tissues and maintaining their homeostasis. Mutations in the genes encoding matrix components cause many serious diseases.

Cells can receive, process, and respond to signals from the external environment because they are equipped with a set of appropriate receptors. The information reaching the receptor, usually in the form of a chemical carrier, may come from the immediate vicinity of the cell and distant parts of the body. The binding of the ligand to the receptor initiates the signalling pathway. The effector response depends on the cell type and the receptor type. A specific group of receptors are adhesion molecules involved in forming cell–cell and cell–matrix connections.

Controlled processes of degradation and secretion of intercellular matrix components mean that tissues are constantly remodelled. However, the body cannot repair certain structural defects on its own. It then needs support in the form of an artificial scaffold on which the cells can settle, multiply and differentiate, and begin to produce the building blocks of the matrix. Over time, the implanted substitute degrades and gives way to a reconstituted protein–polysaccharide network. When designing materials for such implants, the influence of hydrophilicity, topography, roughness, and surface functional groups on cell growth processes must be considered. The results of studies on biomaterials do not give conclusive results, but it should be kept in mind that they are highly dependent on the cell type.

## Figures and Tables

**Figure 1 cells-11-00914-f001:**
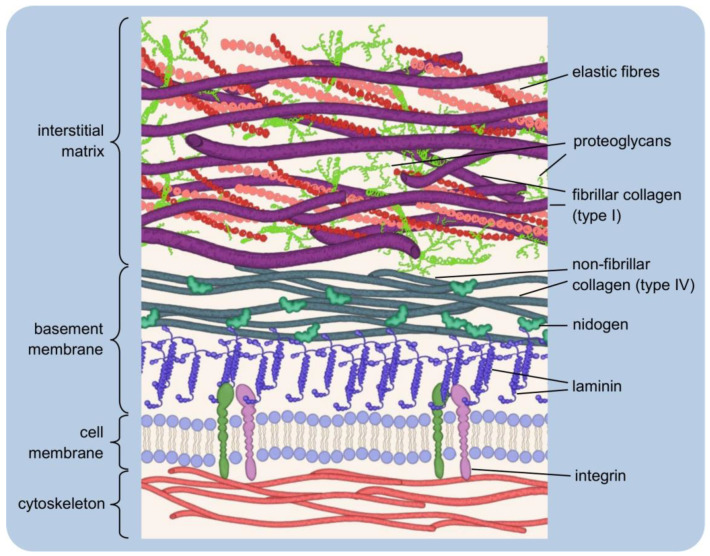
Simplified extracellular matrix structure: three-dimensional macromolecular network composed of various proteins and polysaccharides. The pericellular matrix forms a layer adjacent to the cells: integrins bind to polymerised laminin, which, in turn, is connected via nidogen to the type IV collagen. Interstitial matrix forms porous network of fibrillar collagens, elastic fibres, and proteoglycans.

**Figure 2 cells-11-00914-f002:**
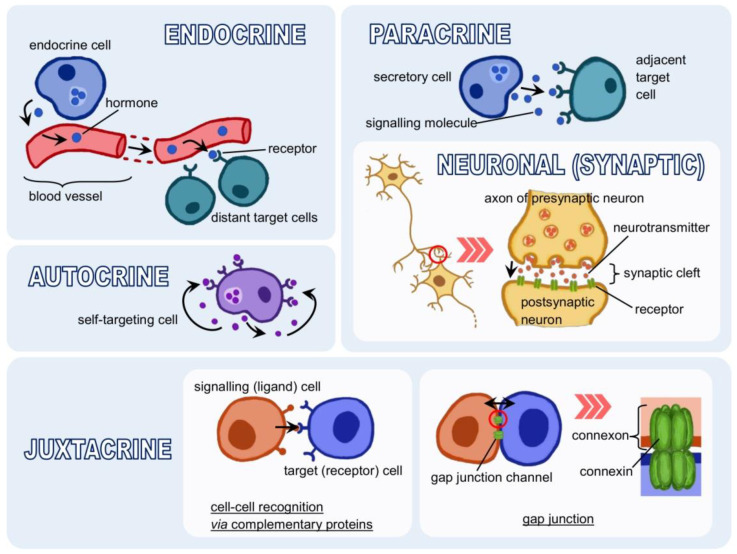
Forms of cell signalling: indirect (endocrine, paracrine, autocrine) and direct (juxtacrine).

**Figure 3 cells-11-00914-f003:**
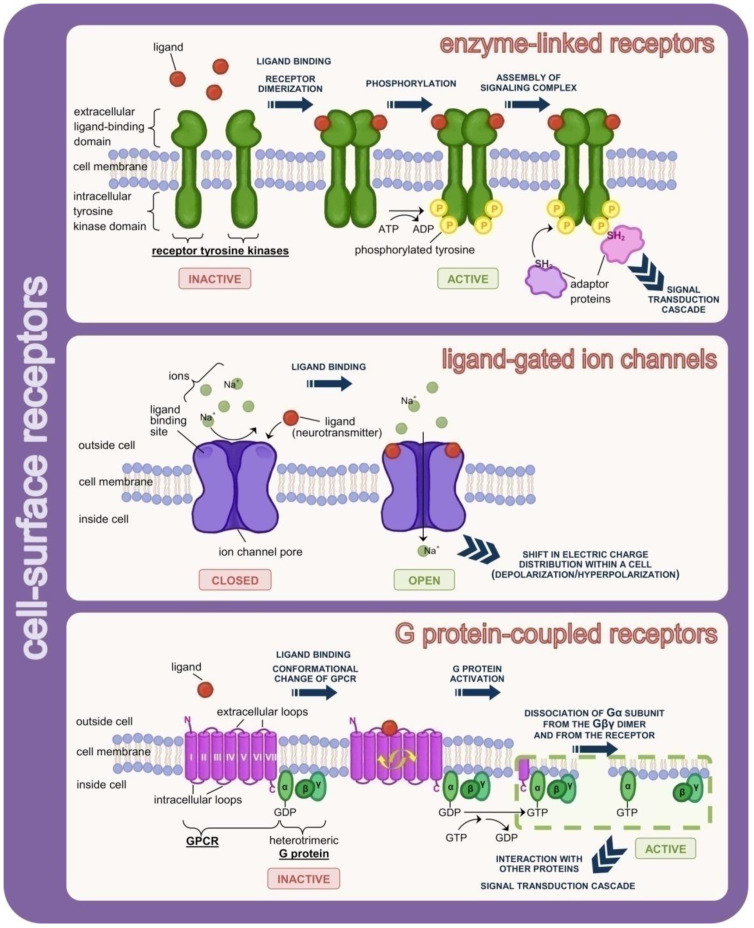
Cell surface receptors embedded in the cell membrane. They act by ligand binding to the extracellular domain of the receptor. The intracellular (cytoplasmic) domain of the receptor communicates via interactions with effector proteins.

**Figure 4 cells-11-00914-f004:**
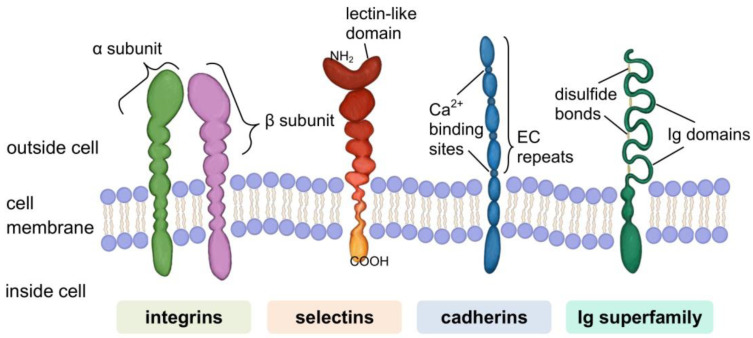
Four families of cell adhesion molecules (CAMs). They are involved in the binding of cells with other cells or with the extracellular matrix in a process called cell adhesion.

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
