# Peer review of "Into the Tissues: Extracellular Matrix and Its Artificial Substitutes: Cell Signalling Mechanisms"

_cells, 2022, doi:10.3390/cells11050914_

Round 1

Reviewer 1 Report

This is a review on the extracellular matrix and the mechanisms its components interacts with the cells. I enjoy reading the manuscript and understand it will be interesting for the journal's readers too. So I recommend it for publication after minor revision. I suggest the authors add a small section on hyaluronan and its interaction mechanisms.

Author Response

  1. As suggested, a small section on hyaluronan and its interaction mechanisms was added with appropriate references. 

Reviewer 2 Report

On the whole, this review is well-structured, well-organized, and enjoyable to read. Though I did not notice any serious flaws, there are a few points that should be rectified--

  1. Graphical abstract: in the cell adhesion section, it should be matrix--  cell membrane ---cell
  2. There is no citation/reference in the introduction section.
  3. Check all the abbreviations used in the manuscript (e.g. extracellular matrix)
  4. References should be rearranged as per the journal format.
  5. Authors should decrease the number of references by omitting older ones and retaining only the most relevant and latest ones, as more recent references are very few.

Author Response

  1. The graphical abstract in the cell adhesion section was corrected to be more precise. 
  2. The number of references was shortened by omitting older and insignificant ones.  
  3. Some references were added to the introduction section.  
  4. The changes made do not decrease the value of this work.